# Effectiveness of Resistance Training Program on Body Composition in Adults Following Vegan Diet versus Omnivorous Diet; Developed in Mobile Health Modality

**DOI:** 10.3390/nu16152539

**Published:** 2024-08-02

**Authors:** Richar Cárcamo-Regla, Rafael Zapata-Lamana, Carolina Ochoa-Rosales, Miquel Martorell, Fernanda Carrasco-Marín, Guillermo Molina-Recio

**Affiliations:** 1Centro de Vida Saludable, Universidad de Concepción, Concepción 4070386, Chile; ricarcamo@udec.cl (R.C.-R.); mmartorell@udec.cl (M.M.); fercarrasco@udec.cl (F.C.-M.); 2Escuela de Kinesiología, Facultad de Salud, Universidad Santo Tomás, Los Ángeles 4440000, Chile; 3Escuela de Educación, Campus Los Ángeles, Universidad de Concepción, Los Ángeles 4440000, Chile; 4Latin American Brain Health Institute (BrainLat), Adolfo Ibáñez University, Santiago 7941169, Chile; caroochoa.r@gmail.com; 5Department of Human Genetics, Radboud University Medical Center, 6525 GA Nijmegen, The Netherlands; 6Departamento de Nutrición y Dietética, Facultad de Farmacia, Universidad de Concepción, Concepción 4070386, Chile; 7Department of Nursing, Pharmacology, and Physiotherapy, University of Córdoba, 14014 Córdoba, Spain; gmrsurf75@gmail.com; 8Lifestyles, Innovation, and Health (GA-16), Maimonides Institute for Biomedical Research of Córdoba (IMIBIC), 14004 Córdoba, Spain

**Keywords:** vegan diet, non-randomized controlled trial, body composition, resistance training, mobile health, fat mass reduction

## Abstract

Background: The vegan diet (VEG_D_) has gained popularity in recent years for ecological and ethical reasons, as well as for its health benefits. In addition to the type of diet, the resistance training program (RT_P_) plays a fundamental role as one of the main natural anabolic stimuli to increase musculoskeletal mass and reduce fat mass. Methods: The study was a 16-week non-randomized controlled clinical trial consisting of three RT_P_ sessions per week. The sample included 70 Chilean individuals, aged between 18 and 59 years, who had been following a VEG_D_ or omnivorous diet (OMN_D_) for the past 6 months. Four groups were established: Vegan Diet Resistance Training Program (VEG_D_-RT_P_), Vegan Diet Control (VEG_D_-C), Omnivorous Diet Resistance Training Program (OMN_D_-RT_P_), and Omnivorous Diet Control (OMN_D_-C). Results: The sample consisted of 47 women and 23 men, with a mean age of 30.1 (±8.6) years. A reduction of 1.20% in the percentage of fat mass (%FM) was observed in the VEG_D_-RT_P_ group (r = 0.554, *p* = 0.016), as well as a reduction of 0.70 kg in kilograms of fat mass (KFM) (r = 0.480, *p* = 0.036). The OMN_D_-RT_P_ group decreased %FM by 0.90% (r = 0.210, *p* = 0.432) and KFM by 0.50 kg (r = 0.109, *p* = 0.683). Conclusions: RT_P_ combined with VEG_D_ or OMN_D_ significantly reduced the percentage of fat mass, although its effect was more significant in the VEG_D_-RT_P_ participants.

## 1. Introduction

The vegan diet (VEG_D_) is characterized by the complete exclusion of animal products. It is primarily composed of plant-based foods such as cereals, legumes, fruits, and vegetables [1,2]. This dietary pattern has gained popularity in recent years due to ecological and ethical motivations and the health benefits it has demonstrated [3,4,5]. The VEG_D_ is rich in dietary fiber, has low energy density, and is low in fats, which can increase the thermic effect of food, a physiological phenomenon that represents approximately 10% of total energy expenditure [6]. Plant-based foods are also rich sources of phytochemicals, vitamins, minerals, antioxidants, and unsaturated fatty acids [7,8,9,10], which contribute to effective glycogen storage and reduced oxidative stress, promoting a reduction in body fat [11].

However, a poorly planned VEG_D_ can lead to micronutrient deficiencies such as vitamin B12, vitamin D, iron, zinc, calcium, and iodine, as well as macronutrient deficits, especially proteins [12,13,14]. Nutritional deficiency has been shown to cause muscle mass loss and bone demineralization in the medium and long term [15]. It is crucial to consider that the musculoskeletal tissue plays a fundamental role in maintaining muscle mass and strength throughout life, providing the contractile force necessary for postures and locomotor activities such as walking and gripping [15]. Additionally, muscle tissue is vital in endocrine regulation, affecting insulin sensitivity and the metabolism of various energy substrates and promoting the maintenance of a healthy body composition [14,16].

Nevertheless, maintaining muscle mass does not rely exclusively on diet [3]. The anabolic effects of regular resistance training (RT_P_) have been demonstrated to be capable of generating a positive protein balance post-exercise through increased muscle protein synthesis, thus acting as a significant natural anabolic stimulus, leading to increased musculoskeletal mass [17]. However, some authors question the anabolic properties of plant-based proteins, mainly due to differences in the composition of specific essential amino acids, such as lysine and methionine, arguing that their availability is lower compared to animal proteins. Despite this, there is insufficient information about a diminished anabolic response in individuals with VEG_D_ compared to those with OMN_D_, as initial studies have not found a clear difference between an omnivorous and a vegan diet in terms of muscle strength or mass [3].

On the other hand, despite the evident benefits of physical exercise, its regular practice has been a challenge for many people who spend prolonged periods in sedentary behavior. Many fail to meet the physical activity recommendations established by the WHO, which suggests 150 to 300 min of moderate to vigorous physical activity per week, or 75 min of vigorous physical activity, including muscle-strengthening activities at least 3 days a week while focusing on major muscle groups [18]. However, the adoption of this sedentary behavior has shown a significant increase in recent years, as observed in the Chilean population where sedentary levels reach 86.7%, resulting in 74.2% of adults being overweight [19].

Given this emerging trend motivating more individuals to adopt a vegan diet (VEG_D_) and the importance that muscle training has demonstrated, it seems pertinent to investigate how RTP can affect body composition when combined with a VEG_D_. Therefore, the aim of this study was to evaluate the effectiveness of RT_P_ implemented through a Mobile Health Modality in individuals who follow a VEG_D_ compared to those who follow an omnivorous diet (OMN_D_), analyzing its effect on body composition (body fat percentage, fat mass, and lean mass). The null hypothesis (H_0_) states that there are no significant differences in body composition between individuals following a vegan diet and those following an omnivorous diet when both groups perform RT_P_, while the alternative hypothesis (H_1_) suggests that there are significant differences in body composition between these groups under the same conditions.

## 2. Materials and Methods

### 2.1. Study Design

A non-randomized controlled clinical trial was conducted with measurements taken at the beginning and end of a 16-week intervention, following the recommendations of TREND (Transparent Reporting of Evaluations with Non-Randomized Designs). Four groups were established: VEG_D_-RT_P_ (22 individuals following a VEG_D_ pattern and participating in RT_P_), VEG_D_-RT_P_ Control (27 participants, VEG_D_ without RT_P_), OMN_D_-RT_P_ (16 subjects, OMN_D_ with RT_P_), and OMN_D_-RT_P_ Control (18 individuals, OMN_D_ without RT_P_). Group assignment was not randomized, with each participant, classified by diet type, choosing to participate in RT_P_ or not based on their convenience. Participant recruitment was carried out in accordance with the principles of the Declaration of Helsinki and was approved by the Ethics, Bioethics, and Biosafety Committee of the University of Concepción (CEBB 1068-2021). Additionally, the study was registered in clinical trials with the identification number NCT06423209. To comply with safety in the use of masks and data management, volunteers were assigned an identification number (ID) upon registration in the form, so neither the evaluator, the nutritionist verifying the diet type, nor the researcher analyzing the data knew to which group each participant belonged.

### 2.2. Participants

The sample consisted of 83 subjects aged 18 to 59 residing in the Biobío and Araucanía regions of Chile who had maintained a VEG_D_ or OMN_D_ for at least the past 6 months. Participant selection was carried out using a non-probabilistic snowball method based on an open invitation disseminated on social media and among VEG_D_ communities from December 2021 to April 2022. Posters with a QR code were used, which redirected people to an online form for the researcher to contact them.

The sample size was calculated based on a t-test for a multiple fixed-effects linear regression. A conservative effect size was chosen for differences in fat mass percentage (f^2^ = 0.15) with 3 study predictors. A statistical power of 80% (β = 0.2) and α = 0.05 were considered. Consequently, the sample size resulted in 55 participants. Additionally, a potential follow-up loss of 45% was considered, thus recruiting 80 subjects, with 20 individuals per study group. This high prediction of follow-up loss was made due to the intervention being in mHealth mode, with limited adherence evidence, potentially leading to higher dropout rates. Power calculations were performed using G*Power version 3.1.9.7.

All baseline assessments were conducted during April and May 2022. Additionally, to verify the participants’ diet type prior to the intervention, a registered dietitian conducted an interview using a validated survey on quantified food consumption trends, involving 14 food groups [20]. Participants were instructed not to make any changes to their diet during the intervention period. All assessments were repeated in October 2022, following the 16-week intervention, under the same protocol conditions, using calibrated instruments and the same evaluator. Furthermore, a brief semi-structured interview was conducted to verify that participants had not made any dietary modifications during this period.

Exclusion criteria included individuals diagnosed with diabetes, uncontrolled hypertension (with Systolic Blood Pressure > 150 and/or Diastolic Blood Pressure > 90 mmHg), those undergoing cancer treatment, chronic kidney disease, or those reporting health issues in a pre-exercise aptitude questionnaire (PAR-Q) [21]. These measurements were performed by a trained and experienced physical education teacher. Additionally, individuals reporting participation in an exercise program or more than 3 h of vigorous exercise weekly were excluded, verified through a self-reported physical activity and sedentary time questionnaire for the last 7 days, using the abbreviated version of the International Physical Activity Questionnaire (IPAQ) [22]. 

### 2.3. Intervention

Assignment to the RT_P_ or Control groups was carried out at the time of completing the intake evaluation, where each participant, according to their type of diet, indicated whether they wished to participate in the RT_P_. Once assigned to a group, the participant could not modify their choice under the intention-to-treat principle. The RT_P_ was implemented remotely between June and September 2022, using the mobile health (mHealth) modality, defined as “the use of mobile and computing technologies in the provision of health and public health services,” through the University of Concepción’s mHealth digital platform (Apptivate, DDI 2023-A-739, Concepción, Chile) [23]. The VEG_D_-RT_P_ (*n* = 22) and OMN_D_-RT_P_ (*n* = 16) groups participated in an RT_P_ for a period of 16 weeks, completing a total of 50 training sessions distributed over three weekly sessions, each lasting between 40 and 50 min.

All participants were asked not to make any modifications to their diet, suggesting that they notify the researcher if they attended a consultation with a nutrition professional during this period or made any changes to their eating habits. During the final evaluation, a brief semi-structured interview was conducted to confirm the maintenance of the diet type during the 16 weeks. Additionally, all participants were required to maintain the physical activity levels reported in the IPAQ questionnaire before the study, suggesting that they should not modify their physical activity levels during the intervention period, considering only the incorporation of RT_P_ sessions for the VEG_D_-RT_P_ and OMN_D_-RT_P_ groups. Participants were required to notify the researcher if they joined a systematic physical exercise program, which was verified by repeating the IPAQ questionnaire and conducting a brief semi-structured interview at the end of the intervention.

The design and supervision of the RT_P_ were carried out by a trainer (physical education teacher) and focused on resistance training with bodyweight exercises targeting major muscle groups, such as the upper body, lower body, and abdominal area. The trainer had an administrator profile on the platform that allowed them to upload the training session, review the attendance list, and monitor the time spent in each session. Additionally, the trainer communicated with the participants via phone calls, text messages, or email to provide feedback or resolve questions about the RT_P_.

Participants received a notification on their mobile phones with the schedule of their next training session and a reminder 30 min before the session started. The platform automatically recorded participants’ attendance and the time spent in each RT_P_ session. To perform this, the participants in the VEG_D_-RT_P_ and OMN_D_-RT_P_ groups had to log into their profiles and access the exercise session from their mobile phones, tablets, or computers. The RTP session remained available on the platform for 24 h, allowing participants to perform their training synchronously or asynchronously, reducing the risk of dropout due to time constraints. At the end of each session, participants had the opportunity to evaluate and share their perceived effort during the training, using the Modified Borg Scale from 0 to 10 [24] (Figure 1).

### 2.4. Anthropometric and Body Composition Measures

Body composition variables (% Fat Mass, kg Fat Mass, and Lean Mass) were monitored and recorded throughout the study using multifrequency electrical impedance (Inbody 120, Lookin’Body 3.0 Data Management Software, Seoul, Republic of Korea). Measurements were taken under fasting conditions of at least 4 h. In addition, participants did not practice any physical exercise during the last 12 h, avoided stimulant beverages such as caffeine, and women were not in their menstrual period [25]. Other anthropometric measurements were taken following standardized anthropometry manual recommendations [26]. Height was measured with participants barefoot using a digital stadiometer Inlab (Inbody, Seoul, Republic of Korea). Nutritional status was obtained by calculating the Body Mass Index (BMI kg/m^2^) [24], while a non-elastic tape measure (Seca 201, Hamburg, Germany) was used for waist circumference. These measurements were performed by a trained and experienced physical education teacher.

### 2.5. Statistical Analysis

For the baseline analysis, quantitative variables were presented with means and standard deviations, and qualitative variables with frequencies and percentages. The normality of the variables was verified using the Kolmogorov–Smirnov test with Lilliefors correction. For the bivariate hypothesis contrast, a one-way ANOVA test was conducted for three or more means, and the Games–Howell post hoc test was used for comparisons between groups. In addition, the baseline variables that presented a statistically significant difference in the comparison across all groups were analyzed with an ANCOVA test (See Appendix A). To measure the effectiveness of the treatment, the Wilcoxon test was used, where the medians of the baseline tests were subtracted from the post-intervention medians of each group. The Kruskal–Wallis test adjusted by the Bonferroni correction was used to compare the medians of the 4 groups. The cut-off points established to assess the strength of the association were the following: very weak, from 0 to 0.19; weak, from 0.2 to 0.39; moderate, from 0.4 to 0.59; strong, from 0.6 to 0.79; and very strong, from 0.8 to 1.0 [27]. Multiple adjusted linear regressions were performed to develop predictive formulas that fit the dependent variables studied. These regressions were developed following the backward stepwise regression method, starting with an equation that includes all the independent variables and removing, one by one, the variables with the highest “*p*-value” until reaching a final model with all significant independent variables (*p* < 0.05). The dependent variable considered was the difference in %FM obtained by subtracting the baseline %FM from the final %FM, and the independent variables were those that showed the greatest statistical significance (*p* < 0.05). The age and the group of participants were organized into dummy variables, taking the OMN_D_-C group as a reference. For all tests, a significance level of less than 5% (*p* < 0.05) was established. The analysis was performed with IBM SPSS Statistics version 22.0.

## 3. Results

For data analysis, the Intention-to-Treat approach will be used, where participants will complete the study according to their original assignment regardless of their adherence to the activities. A total of 83 individuals were initially selected to participate in the study, of whom 49 followed a VEG_D_ and 34 an OMN_D_. At the end of the study, thirteen participants did not complete the activities; five of them were participating in RT_P_ and eight participants were in the control group. The main reasons cited by RT_P_ participants for not continuing with the study were a lack of time to attend RT_P_ sessions (three participants) and physical injury (two participants). Both individuals with physical injuries stated that their injuries were unrelated to the intervention activities. Control group participants did not attend the second evaluation, all citing a lack of time as the primary reason. The total sample of participants who completed the study activities consisted of 70 participants, of whom 40 followed a VEG_D_ and 30 followed an OMN_D_. For the management of missing data, participants who did not comply with a minimum of two RTP sessions per week during the intervention or did not attend the final evaluation were not included. Of the evaluated participants, no missing data were present (Figure 2).

### 3.1. Baseline Data

The final sample included 70 participants, with 67.1% being females and 32.9% males. The participants’ average age was 30.1 (±8.6) years. Regarding anthropometric variables, a significant difference in BMI was observed among all groups (*p* < 0.030), with the OMN_D_-C group having an average BMI of 4.46 kg/m^2^ higher than the VEG_D_-RT_P_ group (*p* = 0.023, IC −8.40–−0.53). Concerning body composition variables, a significant difference in KFM was found among all groups (*p* = 0.010), with the greatest difference seen in the OMN_D_-C group, showing an average of 9.30 kg more than the VEG_D_-RT_P_ group (*p* = 0.013, IC −16.91–−1.68). Detailed information on baseline values for study variables is available in Table 1.

### 3.2. Post-Intervention Results

#### 3.2.1. Effect of the Intervention on Variables by Group

When comparing the differences (initial value–final value) for each group after completing the 16-week intervention program, the VEG_D_-RT_P_ group showed a statistically significant reduction in the %FM variable by 1.20%, with a moderate size of response to treatment (r = 0.554, *p* = 0.016), while the OMN_D_-RT_P_ experienced a decrease of 0.90%, with a weak size of response to treatment (r = 0.210, *p* = 0.432). Regarding the KFM variable, the VEG_D_-RT_P_ group experienced a reduction of 0.70 kg, with a moderate size of response to treatment (r = 0.480, *p* = 0.036), while in the OMN_D_-RT_P_ group, this variable decreased by 0.50 kg, with a wake size of response to treatment (r = 0.109, *p* = 0.683).

In the groups that were not part of the RT_P_, a statistically significant increase in Body Weight was observed in the VEG_D_-C by 1.30 kg, with a moderate response size to the absence of treatment (r = 0.414, *p* = 0.058), while in the OMN_D_-C, this increase was higher 2.35 kg, with a moderate response size to the absence of treatment (r = 0.590, *p* = 0.018). Consequently, a significant increase in BMI was also observed in both groups. Additionally, in the Waist Circumference variable, an increase of 1 cm was observed in the VEG_D_-C group, with a moderate response size to the absence of treatment (r = 0.474, *p* = 0.030) and 1 cm in the OMN_D_-C group, with a moderate response size to the absence of treatment (r = 0.563, *p* = 0.024). In body composition variables, the %FM increased by 1.50% in the OMN_D_-C group, with a strong response size to the absence of treatment (r = 0.731 *p* = 0.003), in addition to an increase of 1.65 kg, with a strong response size to the absence of treatment (r = 0.705, *p* = 0.005) in the KFM variable (Table 2).

#### 3.2.2. Comparison of Pre–Post Intervention Differentials Across All Groups

When comparing the differences (initial value–final value) across all groups, statistically significant differences were observed in body composition variables %FM (<0.001), and KFM (*p* < 0.001). There were no significant differences in the other variables, Weight, BMI, Waist Circumference, or KLM (Table 3).

#### 3.2.3. Linear Regression of the Variable % Fat Mass Difference

A linear regression was performed for the %FM outcome variable, obtained by subtracting the baseline %FM from the post-intervention %FM. Using the correlated independent variables, with a coefficient of determination (R^2^) of 0.256 and a significance in the goodness of fit of <0.001, with a 95% confidence interval (CI), it was observed that, keeping all other variables constant, the VEG_D_−RT_P_ group had a %FM that was 3.61% lower (*p* = 0.001, CI −5.22–−2.00), the VEG_D_−C group had a %FM that was 1.85% lower (*p* = 0.024, CI −3.44–−0.26), and the OMN_D_−RT_P_ group had a %FM that was 2.33% lower (*p* = 0.001, CI −4.00–−0.69) compared to the %FM of the OMN_D_−C group (Table 4).

## 4. Discussion

### 4.1. Body Fat Mass

When analyzing the results obtained in the VEG_D_-RT_P_ group, a statistically significant decrease of 1.20% was observed in the %FM variable (r = 0.554, *p* = 0.016) and 0.7 kg in the KFM variable (r = 0.480, *p* = 0.036). It has been widely reported that a VEG_D_ generates a positive impact on the reduction in fat mass because plant-based foods have a lower fat content and are high in dietary fiber [1,5,6]. These characteristics reduce the energy density of meals, generating changes in the mitochondrial activity of muscle cells [28]. Furthermore, the high antioxidant effect of plant-based foods has been proven, thanks to higher levels of vitamin C, vitamin E, and beta-carotene, as well as greater production of antioxidant enzymes [11]. In addition to the type of diet, benefits have been reported when combining a VEG_D_ with physical exercise, especially RT_P_, showing effects on reducing fat mass and increasing musculoskeletal mass [17]. It is possible that the aforementioned antioxidant disposition of VEG_D_ is a determining factor in the decrease in fat mass observed in the VEG_D_-RT_P_ group due to an enhancement with physical exercise. A study conducted by Boutros et al. evaluated women who had maintained a VEG_D_ or omnivorous diet for the last 2 years and who also reported being physically active, finding a lower BMI and %FM in women with VEG_D_ [27]. However, unlike our research, this study was not a clinical trial, and the participants reported their physical activity through questionnaires, which makes it difficult to accurately measure the influence of physical exercise on their results.

On the other hand, the participants in the OMN_D_-RT_P_ group experienced a slight decrease of 0.90% in the %FM variable (r = 0.210, *p* = 0.432) and 0.50 kg in the KFM variable (r = 0.109, *p* = 0.683), although these reductions did not reach statistical significance. It is likely that the reduced effect observed in the OMN_D_-RT_P_ participants is due to the fact that Western diets rich in animal products induce a high acid load, which is not adequately compensated with dietary buffers such as fruits and vegetables, leading to chronic metabolic acidosis [29,30]. Therefore, considering the eating habits of the Chilean population, characterized by high consumption of meat, ultra-processed foods, and low consumption of fruits and vegetables [19], these results suggest that RT_P_ generates benefits in the body composition of individuals with OMN_D_. However, a stimulus of two to three sessions per week may be insufficient to achieve a significant reduction in fat mass, especially since, in our study, participants were advised not to make dietary changes during the intervention period.

When analyzing the results of the participants in the control groups, we observe that the OMN_D_-C group had a statistically significant increase of 2.35 kg in body weight (r = 0.590, *p* = 0.018), 0.85 kg/m^2^ in BMI (r = 0.554, *p* = 0.027), and 1 cm in waist circumference (r = 0.563, *p* = 0.024), reflecting an increase of 1.50% in the %FM variable (r = 0.731, *p* = 0.003) and 1.65 kg in the KFM variable (r = 0.705, *p* = 0.005). The VEG_D_-C group also increased their body weight by 1.30 kg (r = 0.414, *p* = 0.058), 0.50 kg/m^2^ in BMI (r = 0.418, *p* = 0.056), and 1 cm in waist circumference (r = 0.474, *p* = 0.030). These results may not be surprising and should be largely related to data provided by official surveys in Chile, which indicate high levels of physical inactivity, reaching 86.7% in the adult population [19]. It is known that maintaining low levels of physical activity is related to lower energy expenditure [31], and its explanation is based on the interaction between body levels of fat mass and lean mass. This interaction originates from the fact that adipose tissue behaves like an endocrine organ capable of releasing metabolic mediators [32]. Therefore, given the existing background in the literature, we can identify the importance of RT_P_ in body composition, consistent with the results obtained in this study on the effect caused by the regular practice of RT_P_ conducted in the intervention of the VEG_D_-RT_P_ and OMN_D_-RT_P_ groups, reflected in the reduction in fat mass in both groups. However, the results obtained by the VEG_D_-RT_P_ participants achieved a greater reduction compared to the OMN_D_-RT_P_ participants.

### 4.2. Musculoskeletal Mass

Therefore, the increase in fat mass levels due to a sedentary lifestyle becomes a factor that also affects a decrease in lean mass, reducing the main anabolic hormones that increase nitrogen retention at the muscular level and favoring the degradation of muscle proteins through an inflammatory state [32]. When evaluating the impact of RT_P_ on muscle mass, the VEG_D_-RT_P_ group showed an increase of 0.40 kg in the KLM variable (r = 0.365, *p* = 0.111), while the OMN_D_-RT_P_ group showed a slight increase of 0.25 kg (r = 0.357, *p* = 0.182), increases insufficient to reach statistical significance. It has been demonstrated that the physiological adaptation of skeletal muscle tissue adjusts as a response to contractile activity induced by training, which is reflected in changes in contractile proteins and their mitochondrial function. These adaptations are conditioned, among other things, by factors such as the duration of the stimulus, rest time, and load intensity [33]. It is in this latter point where the low response of the VEG_D_-RT_P_ and OMN_D_-RT_P_ groups on the KLM variable can be explained, as the intensity in the training load was developed using bodyweight exercises, which may be an insufficient contractile stimulus to generate greater muscle hypertrophy. There are studies where RT_P_ exercises were incorporated in people with VEG_D_ and OMN_D_, showing effects on muscle mass. This is demonstrated in the study by Hevia Larraín et al., which reported changes in lean mass in participants with VEG_D_ and OMN_D_ after a 12-week intervention where RT_P_ was developed with 2 to 3 weekly training sessions incorporating overload exercises. Their results showed an increase in muscle mass in participants with both types of diets, without differences between VEG_D_ and OMN_D_ [34].

The muscle response observed in our research and in other studies to a resistance training regimen [34,35] can be explained by the fact that the contractile stimulation of the muscle increases the sensitivity to the anabolic properties of amino acid or protein administration. The postprandial increase in circulating plasma leucine concentration is of lesser importance when protein is consumed after exercise. Therefore, the lower leucine content of most plant-based proteins does not restrict postprandial muscle protein synthesis rates during exercise recovery [14]. A study evaluated healthy and recreationally active individuals, where, through imaging, they examined the effect of a vegan diet and reported that no differences were found in vascular structure and function, nor in the properties of skeletal muscle, between those who follow a habitual vegan diet and omnivores [36]. Moreover, in recent years, some elite athletes have been able to follow a VEG_D_, managing to maintain their athletic performance over time [8]. Therefore, it can be deduced that muscle mass depends on a positive muscle protein balance over time, which can be achieved through adequate intake of proteins and essential amino acids, enhanced with the combination of physical exercise [13].

In summary, it is important to note that no differences were observed in the muscle mass levels of the groups when comparing the different types of diets. These results are relevant due to the widespread belief, manifested by some authors, that a poorly constructed vegan diet could lead to a deficiency of essential amino acids such as leucine, methionine, and lysine, due to lower bioavailability in plant-based foods and, therefore, be related to muscle mass loss. However, it has been noted that the vast majority of these nutrients can be incorporated into the diet complementarily through nutritional supplementation, obtaining a sufficient nutritional load for anabolic stimulation that allows for maintaining muscle mass levels within normal ranges [14].

### 4.3. Limitations of the Study

Some important design aspects and limitations of this study require further consideration and context. One of these is the non-random distribution of participants into groups, which affects the homogeneity of the sample. Significant differences were observed between groups in variables such as age and BMI, which could introduce bias into the analysis of results. The choice of this distribution type was necessary to address difficulties encountered in achieving the required number of participants in each group. However, ANCOVA analysis indicated that no variation in results was observed due to baseline differences, suggesting that our study results would not be significantly affected by this disparity. Additionally, our adjusted linear regression model showed that independent variables did not affect the observed changes post-intervention, thereby reducing the risk of bias in results analysis. Despite these findings, the baseline difference caused by non-randomized distribution may still pose a limitation, suggesting that future studies should consider random assignment. Furthermore, although age did not exert a statistically significant influence, it was included in the multiple linear regression model, adjusted using dummy variables separating participants by groups. This decision was based on literature indicating changes in body composition with advancing age [37]. This effect may not be observed in our results due to the distribution and size of our sample.

Since participants were not subjected to nutritional intervention, the information on dietary intake may be imprecise due to the study’s duration. Dietary records may have been compromised by a lack of information or recall errors. One way to reduce this bias would have been to repeat the food frequency and consumption questionnaire in weeks 8 and 16. However, this practice is impractical due to the high time demands on both professionals and participants. Nevertheless, to mitigate this bias, a semi-structured interview was incorporated in week 16, with general but relevant questions, such as whether participants had attended nutritional consultations during the past 16 weeks, started any dietary regimen, or made modifications to their diet during this period. Therefore, it is recommended that future research includes more concise forms of dietary habit recording or self-reporting, which, although they may not be as precise as a food frequency and consumption questionnaire, allow for more frequent evaluations during the study.

Although the characteristics of the RTP applied in this study align with the recommendations of the American College of Sports Medicine (ACSM) [35], it would be interesting to develop a program with similar characteristics, increasing the workload and incorporating overload exercises to enhance muscle contractile stimulation. It is noteworthy that adherence to the RTP was 87%, suggesting that it is a training protocol that can be sustained over time.

Finally, the segmentation of the age group of VEG_D_ participants ranged from 20 to 36 years. However, it should be noted that this type of diet is relatively new and emerging, making it difficult to find older participants, raising the question of whether the effects of VEG_D_ or the benefits of RT_P_ on body composition could be maintained in the following years of their life cycle [32]. However, the authors are convinced that these findings contribute to a body of evidence that values the importance of active lifestyles, especially the benefits of RT_P_, regardless of diet type. Additionally, according to their findings, a well-designed and supplemented plant-based diet would not negatively affect musculoskeletal mass, and on the contrary, its positive effects are enhanced when physical activity is incorporated. This opens the door to future research with a larger number of participants, randomized intervention assignments, and longer follow-up periods to verify effectiveness over time. Another limitation could be the segmentation of the age group of VEG_D_ participants, ranging from 20 to 36 years. However, it should be noted that this type of diet is relatively new and emerging, making it difficult to find older participants, raising the question of whether the effects of VEG_D_ or the benefits of RT_P_ on body composition could be maintained in the following years of their life cycle.

## 5. Conclusions

The results of this study indicate that the VEG_D_-RT_P_ participants experienced a reduction in body fat, showing superior results compared to those in the OMN_D_-RT_P_. On the other hand, both control groups showed an increase in weight, BMI, and fat mass, presumably attributable to adopting a sedentary behavior and dietary modifications during the intervention period. Although the vegan diet has been associated with weight reduction and the maintenance of a healthier body composition, combining it with an active lifestyle and regular physical exercise seems to be a more efficient strategy for reducing fat mass. Additionally, the sample of participants following a VEG_D_ consisted mainly of younger individuals, suggesting the need for more studies in older populations to better understand the long-term effects on body composition when combined with RT_P_.

## Figures and Tables

**Figure 1 nutrients-16-02539-f001:**
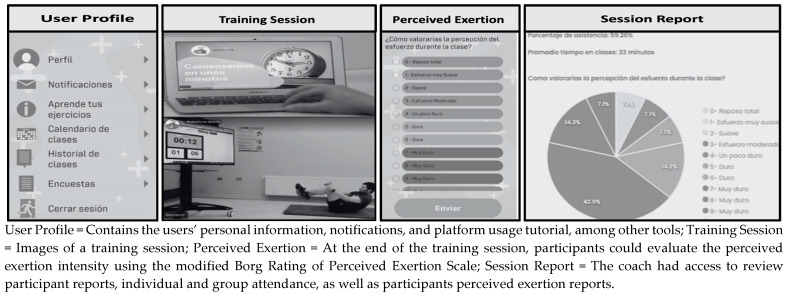
Images extracted from the APPTIVATE platform.

**Figure 2 nutrients-16-02539-f002:**
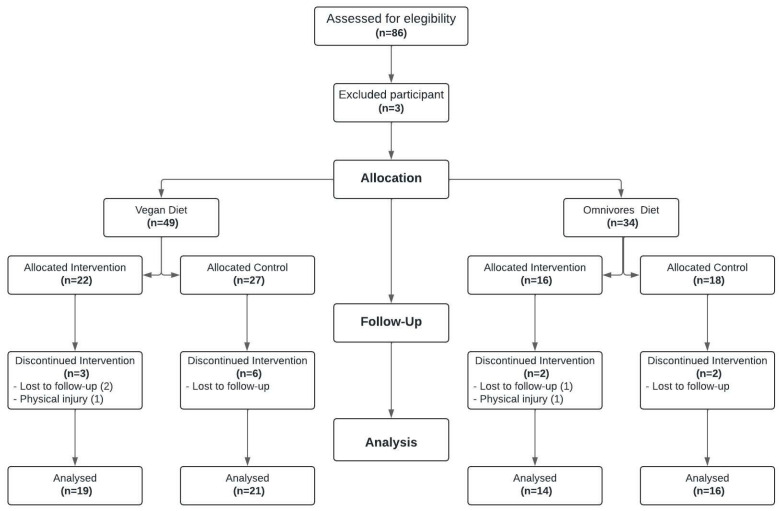
This figure shows the flow diagrams of the participants, separated according to their type of diet, and assigned to either the intervention group with RT_P_ or the control group.

**Table 1 nutrients-16-02539-t001:** Baseline characteristics of participants (*n* = 70).

	Total (*n* = 70)	VEG_D_-RT_P_ (*n* = 19)	VEG_D_-C (*n* = 21)	OMN_D_-RT_P_ (*n* = 14)	OMN_D_-C (*n* = 16)	*p*
Sociodemographics						
Women (*n*, %)	47 (67.1%)	10 (52.6%)	4 (19%)	10 (71.4%)	11 (68.8%)	0.154 ^a^
Males (*n*, %)	23 (32.9%)	9 (47.4%)	17 (81%)	4 (28.6%)	5 (31.2%)	
Age (*Mean*, *SD*)	30.13 (±8.6)	27.53 (±4.1) ^≠ =^	26.81 (±5.0) ^=^	33.10 (±11.0) ^=^	35.00 (±11.1) ^≠ =^	0.030 ^w^
CD (*n*, %) Yes	12 (17.1%)	1 (5.3%)	4 (19%)	4 (28.6%)	3 (18.8%)	0.251 ^w^
CD (*n*, %) No	58 (82%)	18 (94.7%)	17 (581%)	10 (71.4%)	13 (81.3%)	
PNC (*n*, %) Yes	48 (68.6%)	16 (84.2%)	17 (81%)	7 (50%)	8 (50%)	0.173 ^w^
PNC (*n*, %) No	22 (31.4%)	3 (15.8%)	4 (19%)	7 (50%)	8 (50%)	
Anthropometry						
Weight (*Mean*, *SD*)	69.11 (±15.2)	62.00 (±8.8) ^≠ =^	72.51 (±18.6) ^=^	68.34 (±11.9) ^=^	73.80 (±16.7) ^≠ =^	0.080 ^a^
Height (*Mean*, *SD*)	164.72 (±8.1)	164.83 (±8.6)	165.41 (±9.2)	164.00 (±3.6)	164.40 (±7.6)	0.960 ^a^
BMI (*Mean*, *SD*)	25.44 (±5.1)	22.73 (±2.0) ^≠ =^	26.61 (±6.7) ^=^	25.40 (±3.6) ^=^	27.20 (±5.3) ^≠ =^	0.030 ^w^
WC (*Mean*, *SD*)	80.90 (±12.7)	76.45 (±7.6)	80.31 (±12.0)	81.20 (±12.8)	86.60 (±16.7)	0.141 ^w^
%FM (*Mean*, *SD*)	31.10 (±9.6)	26.30 (±9.2)	32.19 (±11.6)	32.20 (±6.4)	34.40 (±7.5)	0.060 ^w^
KFM (*Mean*, *SD*)	22.05 (±10.4)	16.33 (±6.1) ^≠ =^	24.51 (±14.0) ^=^	22.03 (±6.5) ^=^	25.62 (±9.5) ^≠ =^	0.010 ^w^
KLM (*Mean*, *SD*)	26.10 (±5.8)	25.19 (±5.3)	26.58 (±5.8)	25.80 (±5.3)	26.71 (±7.0)	0.850 ^a^

VEG_D_-RT_P_ = Vegan Diet Resistance Training Program; VEG_D_-C = Vegan Diet Control; OMN_D_-RT_P_ = Omnivore Diet Resistance Training Program; OMN_D_-C = Omnivore Control; CD = Chronic diseases; PNC = Previous nutritional control; BMI = Body Mass Index; WC = Waist circumference; %FM = % Fat Mass; KFM = Kilograms Fat Mass; KLM = Kilograms Lean Mass; SD = Estándar Deviation; ^a^ = ANOVA; ^W^ = Wells; ^≠ =^ = Significant differences identified between the groups according to the post hoc test; (Games–Howell). ^=^ = Significant differences identified between the groups according to the post hoc test.

**Table 2 nutrients-16-02539-t002:** Effect of the Intervention on Variables by Group.

	Basal	Post-I	Effect Size (r)	*p*
	VEG_D_-RT_P_-I (*n* = 19)			
Weight	61.40 [53.10; 70.20]	59.90 [55.60; 67.10]	0.092	0.687
BMI	23.00 [21.40; 24.00]	22.70 [20.90; 23.40]	0.179	0.434
WC	78.00 [71.50; 80.50]	74.00 [72.00; 80.00]	0.336	0.143
%FM	26.60 [19.00; 33.50]	26.50 [14.40; 33.80]	0.554	0.016 *
KFM	16.30 [10.30; 21.60]	15.40 [7.90; 20.20]	0.480	0.036 *
KLM	25.10 [21.10; 29.60]	24.70 [21.00; 29.70]	0.365	0.111
	VEG_D_-C (*n* = 21)			
Weight	68.80 [60.20; 80.60]	70.00 [59.40; 82.55]	0.414	0.058 *
BMI	25.10 [21.30; 31.35]	24.60 [21.85; 31.65]	0.418	0.056 *
WC	77.00 [69.50; 89.50]	77.50 [70.25; 91.00]	0.474	0.030 *
%FM	29.60 [21.95; 43.15]	29.00 [24.10; 43.25]	0.023	0.917
KFM	21.30 [14.40; 32.60]	19.90 [15.15; 33.40]	0.330	0.130
KLM	24.20 [22.50; 31.25]	25.40 [22.90; 31.15]	0.376	0.085
	OMN_D_-RT_P_ (*n* = 14)			
Weight	67.40 [59.78; 73.30]	68.60 [59.60; 74.98]	0.178	0.506
BMI	24.50 [22.50; 27.10]	25.10 [22.80; 27.78]	0.310	0.247
WC	78.50 [73.38; 84.25]	79.00 [75.00; 84.00]	0.118	0.658
%FM	33.55 [27.88; 36.95]	33.70 [27.80; 35.88]	0.210	0.432
KFM	20.05 [17.75; 24.03]	20.50 [15.95; 27.30]	0.109	0.683
KLM	24.75 [21.00; 30.93]	24.75 [21.18; 30.85]	0.357	0.182
	OMN_D_-C (*n* = 16)			
Weight	67.70 [61.23; 88.3]	71.20 [62.53; 91.75]	0.590	0.018 *
BMI	26.1 [23.78; 30.60]	27.05 [25.03; 31.28]	0.554	0.027 *
WC	81.75 [72.63; 104.63]	82.75 [75.50; 100.50]	0.563	0.024 *
%FM	33.05 [27.98; 39.95]	34.80 [28.68; 43.60]	0.731	0.003 *
KFM	24.4 [19.2; 27.6]	26.65 [21.36; 30.25]	0.705	0.005 *
KLM	24.3 [21.2; 31.7]	24.50 [20.63; 34.50	0.117	0.640

VEG_D_-RT_P_ = Vegan Diet Resistance Training Program; VEG_D_-C = Vegan Diet Control; OMN_D_-I = Omnivore Diet Resistance Training Program; OMN_D_-C = Omnivore Control; BMI = Body Mass Index; WC = Waist circumference; %FM = % Fat Mass; KFM = Kilograms Fat Mass; KLM = Kilograms Lean Mass; Basal = Median and Interquartile Range Pre-Intervention Assessment; Post-I = Median and Interquartile Range Post-Intervention Result; * = Statistical Significance < 0.05.

**Table 3 nutrients-16-02539-t003:** Comparison of pre–post intervention differentials across all groups.

	VEG_D_-RT_P_ (*n* = 19)		VEG_D_-C (*n* = 21)		OMN_D_-RT_P_ (*n* = 14)		OMN_D_-C (*n* = 16)		
	Diferential	r	Diferential	r	Diferential	r	Diferential	r	*p*
Weight	−0.20 [−1.80; 1.90]	0.092	1.30 [−0.40; 2.35]	0.414	0.40 [−0.93; 1.80]	0.178	2.35 [0.27; 3.43]	0.590	0.087
BMI	−0.20 [−0.80; 0.70]	0.179	0.50 [−0.20; 0.95]	0.418	0.15 [−0.23; 0.80]	0.310	0.85 [−0.15; 1.30]	0.554	0.054
WC	−1.00 [3.00; 1.50]	0.336	1.00 [0.50; 2.50]	0.474	1.00 [−1.25; 1.62]	0.118	1.00 [0.50; 2.38]	0.563	0.064
%FM	−1.20 [−4.00; 0.40] ^≠ =^	0.554	−0.20 [−1.55; 1.10] ^=^	0.023	−0.90 [−2.82; 1.45] ^=^	0.210	1.50 [0.35; 3.13] ^≠ =^	0.731	<0.001 *
KFM	−0.70 [−2.80; 0.40] ^≠ =^	0.480	0.60 [−0.55; 2.00] ^=^	0.330	−0.50 [−2.05; 1.40] ^=^	0.109	1.65 [0.95; 3.22] ^≠ =^	0.705	<0.001 *
KLM	0.40 [−0.40; 0.90]	0.365	0.20 [−0.20; 0.90]	0.376	0.25 [−0.22; 0.58]	0.357	−0.30 [−0.90; 0.85]	0.117	0.510

VEG_D_-RT_P_ = Vegan Diet Resistance Training Program; VEG_D_-C = Vegan Diet Control; OMN_D_-RT_P_ = Omnivore Diet Resistance Training Program; OMN_D_-C = Omnivore Control; BMI = Body Mass Index; WC = Waist Circumference; %FM = % Fat Mas; KFM = Kilograms Fat Mass; KLM = Kilograms Lean Mass; Diferential; Median and Confidence Interval Post-Intervention Result—Basal Result; * = Statistical Significance < 0.05; ^≠ =^ = Significant differences between the groups are identified according to the Benferroni Test; ^=^ = Significant differences between the groups are not identified according to the Bonferroni Test; r = Effect Size.

**Table 4 nutrients-16-02539-t004:** Linear Regression of the Variable % Fat Mass Difference.

	(Unadjusted) Analysis				(Adjusted) Analysis			
% Fat Mass Difference	Coef	SE	CI 95%	*p*	Coef	SE	CI 95%	*p*
Age	0.035	0.35	−0.035–0.106	0.322	−0.006	0.034	−0.075–0.063	0.861
VEG_D_-RT_P_	−3.564	0.756	−5−074–−2.054	<0.001 *	−3.610	0.804	−5.220–−2.003	<0.001 *
VEG_D_-C	−1.797	0.740	−3.274–−0.327	0.018 *	−1.850	0.800	−3.440–−0.260	0.024 *
OMN_D_-RT_P_	−2.321	0.816	−3.950–−0.692	0.006 *	−2.33	0.82	−4.00–−0.69	0.001 *

VEG_D_-RT_P_ = Vegan Diet Resistance Training Program; VEG_D_-C = Vegan Diet Control; OMN_D_-RT_P_ = Omnivore Diet Resistance Training Program; OMN_D_-C = Omnivore Control; % Fat Mass Difference = % Fat Mass pre intervention—% Fat Mass post intervention; Coef = Unstandardized beta coefficient; SE = Estándar Error; IC 95% = Confidence Interval 95%; * = Statistical significance < 0.05. (For the dummy variables, the Omnivorous Diet Control Group was taken as the reference variable).

## Data Availability

The data presented in this study are available on request to the corresponding author.

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
