# Peer review of "Effectiveness of Resistance Training Program on Body Composition in Adults Following Vegan Diet versus Omnivorous Diet; Developed in Mobile Health Modality"

_nutrients, 2024, doi:10.3390/nu16152539_

Round 1

Reviewer 1 Report

Comments and Suggestions for Authors

The manuscript addresses an important topic, evaluating the effects of a resistance training program on body composition in adults following vegan and omnivorous diets. While the study is relevant and timely, several areas require clarification and improvement to increase the validity and readability of the paper.

Initial BMI Differences:

Problem: There are significant differences in baseline BMI values ​​between groups.

Recommendation: These differences should be acknowledged in the methodology and discussion sections. It is critical to correct for baseline BMI differences using appropriate statistical methods such as ANCOVA to ensure that results are not affected by these initial differences.

High dropout rate:

Problem: The attrition rate is high, with 13 participants not completing the study.

Recommendation: Provide a detailed discussion on the reasons for the high attrition rate. Discuss any measures taken to minimize dropouts and how missing data were handled in the analysis. It would be useful to use the “intention-to-treat” analysis to include all participants initially enrolled in the study.

Diet management:

Problem: Management and verification of adherence to vegan and omnivore diets are not clearly described.

Recommendation: Include detailed information on how adherence to the diet was monitored and ensured. Specify the tools and methods used for dietary assessments and the frequency of such assessments. This will help understand the consistency and reliability of the dietary intervention.

Discussion Section:

Problem: The discussion is unclear and lacks depth.

Recommendation: The discussion should be refined to clearly summarize the main findings and their implications. Compare your findings to existing literature to provide context. Discuss the potential mechanisms underlying the observed effects and acknowledge the limitations of the study. Highlight the study's contributions to the field and suggest directions for future research.

Statistic analysis:

Problem: The statistical analysis section is missing some details.

Recommendation: Ensure that all statistical methods used are described in detail. This includes how baseline differences were adjusted for and the specific tests used for within-group and between-group comparisons.

Clarity and readability:

Problem: Some sections of the manuscript are difficult to follow.

Recommendation: Make sure your manuscript is clear and concise. Consider asking a colleague to review the manuscript for clarity and readability. Use titles and subtitles to better organize the content.

Detailed comments by section:

Introduction:

The introduction is well written and provides a good foundation. However, it could benefit from a clearer statement of the hypotheses and objectives of the study.

Methods:

Clearly describe the randomization process (if any) and the rationale behind the sample size calculation.

Provide more detail on diet assessment tools and frequency of assessments to ensure dietary adherence.

Results:

Present results with clear tables and figures. Make sure all tables and figures are cited in the text.

Discuss the implications of the attrition rate on the study results.

Discussion:

Refine the discussion to focus on key findings and their implications. Compare your results with those of other studies.

Discuss the potential mechanisms underlying the observed effects.

Acknowledge the limitations of the study, including the high attrition rate and any potential bias.

Conclusion:

Provide a concise summary of the study findings and their implications for practice and future research.

Comments on the Quality of English Language

English is fine

Author Response

Thank you very much for taking the time to review this manuscript in such detail, sharing your knowledge and experience to improve this article. Below, in the point-by-point section, you will find detailed responses with corrections and comments. I hope that this development meets the objective of clarifying your doubts, leading to an improvement in the content of this article.

Reviewer 2 Report

Comments and Suggestions for Authors

Abstract:

·         The results section is presented clearly, with specific data on the changes in fat mass for both the VEGD-RTP and OMND-RTP groups. However, the statistical significance of the findings should be emphasized more. For example, while the p-values are provided, the practical significance of these changes in fat mass should be discussed briefly. Furthermore, the results for the control groups (VEGD-C and OMND-C) are mentioned but not detailed, which could lead to an incomplete understanding of the study's findings. Including a comparative analysis between the intervention and control groups would provide a more comprehensive overview.

·         The keywords are appropriate and relevant to the study. However, including terms like "non-randomized controlled trial," "body composition," and "fat mass reduction" would improve the searchability and relevance of the abstract.

1. Introduction:

·         While the introduction provides a broad overview of the vegan diet and physical activity, it sometimes relies too heavily on general statements. More specific details about previous studies and their findings could strengthen the argument.

·         The introduction mentions various benefits and drawbacks of the VEGD but lacks a critical analysis of the existing literature.

·         The section would benefit from a more explicit identification of the research gaps that this study aims to address.

·         The introduction appears to favor VEGD without equally addressing the benefits of omnivorous diets.

·         While the introduction provides relevant background information, it could more explicitly connect this information to the specific objectives and hypotheses of the study.

2. Materials and Methods

·         The study design, described as a 16-week non-randomized controlled clinical trial, is appropriate for investigating the impact of an RTP on body composition in adults following different diets. The adherence to TREND  guidelines strengthens the methodological rigor. However, non-randomized allocation introduces potential selection bias, which should be acknowledged as a limitation. Clarifying why a non-randomized design was chosen and how this may impact the results would benefit readers.

·         The detailed recruitment process and ethical considerations are commendable, but additional strategies to enhance retention should be explored.

·         Providing various communication channels for participants to interact with trainers is a positive aspect, ensuring support and potentially enhancing adherence. However, the reliance on self-reported effort and adherence through the platform might introduce reporting bias.

·         Multifrequency electrical impedance to monitor body composition is appropriate and provides reliable data on % Fat Mass, kg Fat Mass, and Lean Mass. The standardized measurement conditions (e.g., fasting state, avoidance of physical activity, and stimulants) add to the robustness of the data collection process. Including additional measures like dietary intake and physical activity levels outside of the intervention period could provide a more comprehensive understanding of the factors influencing body composition changes.

3. Results:

·         The baseline characteristics are clearly described, and significant group differences are identified. However, the explanation for the choice of non-randomized design and its potential impact on baseline differences could be elaborated upon.

·         The statistical significance is indicated, but there is a need for a more detailed interpretation of non-significant findings.

·         The tables and figures are informative, but some values (e.g., confidence intervals) could be better highlighted to emphasize the statistical reliability of the findings.

·         The use of p-values is appropriate, but effect sizes should also be reported to provide a more complete picture of the practical significance of the results.

4. Discussion:

·         The discussion lacks a deep analysis of the mechanisms underlying the observed differences between the VEGD-RTP and OMND-RTP groups. While the authors mention the lower fat content and higher dietary fiber of the VEGD, a more detailed exploration of biochemical and physiological mechanisms is needed.

·         The comparison between the intervention and control groups is somewhat cursory. A more detailed comparative analysis, highlighting the statistical significance and the practical significance of the findings, is required. Discussing effect sizes and their implications for real-world applications would be valuable.

·         The role of nutritional supplementation in the VEGD-RTP group is acknowledged but not thoroughly explored. More details should be included on the types and dosages of supplements used, their potential impacts on the results, and how they were controlled in the study design.

·         The non-significant findings in some measures (e.g., KLM) are briefly mentioned but not sufficiently discussed. Potential reasons for these non-significant changes, such as the nature of the resistance training program (bodyweight exercises) and its intensity and duration, should be explored.

·         Some discussion sections are repetitive and could be streamlined for clarity. Consolidating these points into a single, concise paragraph would improve readability.

·         The authors should elaborate on how the dietary fiber in VEGD specifically impacts body fat reduction, possibly citing studies on metabolic rate and energy expenditure.

·         The increase in body weight and BMI in control groups should be discussed in relation to potential behavioral changes or seasonal variations that could have influenced these outcomes.

·         The mention of bodyweight exercises not providing enough load for hypertrophy is crucial. The authors could suggest alternative training protocols that might yield different results.

Comments on the Quality of English Language

The language is generally clear and understandable, with only occasional minor errors that do not significantly impede comprehension.

Author Response

Estimado revisor,

Muchas gracias por tomarse el tiempo de revisar este manuscrito con tanto detalle, compartiendo sus conocimientos y experiencia para enriquecer este artículo. A continuación, en la sección punto por punto, encontrará respuestas detalladas con correcciones y comentarios. Espero que este avance resuelva sus inquietudes y contribuya a mejorar el contenido de este artículo.

Atentamente,

Round 2

Reviewer 1 Report

Comments and Suggestions for Authors

The authors have answered to some of my previous comments. However there are still some major shortcomings to be fixed.

BMI differences and impact on outcomes: The manuscript acknowledges significant differences in BMI at baseline between the groups, with the OMND-C group having a higher mean BMI than the VEGD-RTP group (p=0.030). These differences could influence the results of the study, in particular the changes in body composition. Although the authors mentioned these differences, further discussion is needed on how these differences were controlled for in the statistical analysis. The manuscript should also explore how these basic disparities could influence the interpretation of the results.

Revision of the abstract: The updated abstract still lacks emphasis on the superior results observed in the VEGD-RTP group. 

Methodology Clarifications: The study acknowledges the non-randomized group assignments, which could introduce selection bias. The authors should provide a detailed explanation of how participants were assigned to groups and any measures taken to mitigate bias. The criteria for verifying that participants maintained their diet type throughout the study should be clarified. 

Comments on the Quality of English Language

English is fine

Author Response

Dear Reviewer,

Thank you very much for taking the time to review this manuscript in such detail, sharing your knowledge and experience to enhance this article. Below, in the point-by-point section, you will find detailed responses with corrections and comments. I hope this development addresses your concerns and contributes to improving the content of this article.

Best regards,

Reviewer 2 Report

Comments and Suggestions for Authors

Abstract:

·         The abstract still lacks a discussion on the practical significance of the changes in fat mass. It would be beneficial to mention the real-world implications of the findings briefly.

·         Terms like "non-randomized controlled trial" and "fat mass reduction" are still missing from the keywords, which would improve the searchability and relevance of the abstract.

Introduction:

·         The critical analysis of the existing literature is still somewhat superficial, and identifying specific research gaps could be more explicit.

Materials and Methods:

·         There is limited discussion on additional strategies to enhance participant retention.

·         Further strategies to mitigate the potential reporting bias due to self-reported effort and adherence have not been sufficiently explored.

·         While dietary intake verification is described, considering physical activity levels outside the intervention period would provide a more comprehensive understanding.

Results:

·         Effect sizes are still not mentioned, which would provide a more complete picture of the practical significance of the results.

Discussion:

·         Nutritional Supplementation: The role of nutritional supplementation in the VEGD-RTP group has not been thoroughly explored. More details on supplements' types, dosages, and impacts would be beneficial.

Comments on the Quality of English Language

The text is generally clear and comprehensible, with only a few minor grammatical or stylistic issues that need correction.

Author Response

(The authors gave the same response as above.)
